# Modified Nano-Fe₂O₃-Paraffin Wax for Efficient Photovoltaic/Thermal System in Severe Weather Conditions

Miqdam T. Chaichan [1], Maytham T. Mahdi [1], Hussein A. Kazem [2], Ali H. A. Al-Waeli [3], Mohammed A. Fayad [1], Ahmed A. Al-Amiery [1,4,*], Wan Nor Roslam Wan Isahak [4,*], Abdul Amir H. Kadhum [5] and Mohd S. Takriff [6]

[1] Energy and Renewable Energies Technology Center, University of Technology, Baghdad 10066, Iraq
[2] Faculty of Engineering, Sohar University, P.O. Box 44, Sohar PCI 311, Oman
[3] Engineering Department, American University of Iraq, Sulaimani 46001, Iraq
[4] Department of Chemical and Process Engineering, Faculty of Engineering and Built Environment, Universiti Kebangsaan Malaysia (UKM), Bangi 43600, Selangor, Malaysia
[5] Faculty of Medicine, University of Al-Ameed, Karbala 56001, Iraq
[6] Chemical and Water Desalination Engineering Program, Department of Mechanical & Nuclear Engineering, College of Engineering, University of Sharjah, Sharjah 26666, United Arab Emirates
* Correspondence: dr.ahmed1975@ukm.edu.my (A.A.A.-A.); wannorroslam@ukm.edu.my (W.N.R.W.I.)

**Abstract:** The development of modern photovoltaic thermal systems (PV/T) is one of the most important steps in the application of using solar energy to produce both electricity and heat. Studies have shown that a system consisting of a heat-collecting tank the is most efficient system, in which the phase change materials (PCMs) are mixed with nanoparticles inside the system that are cooled by a cooling fluid (preferably a nanofluid). The PCMs have a high capacity to store energy in the form of latent heat. Nanoparticles are added to PCMs to treat and improve the low thermal conductivity of these materials. In this experimental study, nano-iron oxide III ($Fe_2O_3$) was added to paraffin wax in multiple mass fractions to evaluate the thermophysical changes that can be occur on the wax properties. Four samples of paraffin–nano-$Fe_2O_3$ were prepared with mass fractions of 0.5%, 1%, 2% and 3%, and their thermophysical properties were compared with pure paraffin (without nano additives). The results from this study showed that adding nano-$Fe_2O_3$ at any mass fraction increases the viscosity and density of the product. Thermal conductivity is improved by adding nano-$Fe_2O_3$ to paraffin wax by 10.04%, 57.14%, 76.19%, and 78.57% when adding mass fractions of 0.5%, 1%, 2%, and 3%, respectively. Stability tests showed that the prepared samples have excellent thermal stability (especially for 0.5% and 1% added nano-$Fe_2O_3$) to acceptable level of stability when adding 3% of nano-$Fe_2O_3$. The nano-$Fe_2O_3$ paraffin PV/T system was tested outdoors to ensure its ability to operate in the harshest weather conditions of Baghdad city. The current experimental results indicated clear evidence of the success of the examined nano-PCM.

**Keywords:** photovoltaic thermal; nano-$Fe_2O_3$; mass fraction; thermal conductivity; stability

## 1. Introduction

Non-renewable energy sources have been widely used to meet global energy production requirements for decades [1]. There are many constraints facing the use of non-renewable resources, such as pricing, and economic, political, and environmental issues. The side effects of using non-renewable energy resources in various sectors have led to an increase in concerns regarding its contribution to global warming and climate change, in addition to the clearly evident increase in environmental pollution. Therefore, there is a global effort from the scientific community to gradually reduce the dependence on these sources and replace them with environmentally friendly renewable sources [2,3]. Examples of renewable sources are wind energy, geothermal energy, hydropower, and solar energy [4]. Solar energy is also an important resource to make life possible on Earth, and it is used for various purposes, such as heating water for domestic purposes, and

heating air for thermal comfort purposes (for instance, heating homes during the winter). Additionally, it can be used to generate electricity directly by using photovoltaic modules through converting solar radiation to electricity. Furthermore, solar radiation can be used in special plants to produce steam used in electricity power generators [5]. Perhaps the most important reason that gives preference to the use of solar energy is that it is an energy available around the world, is free and environmentally friendly, and that it does not cause environmental pollution. The disadvantage of solar energy is that it is not available in the absence of the sun and its energy is difficult to store. There are several methods to store solar energy, including electrical and thermal storage [6]. Thermal energy obtained from the photovoltaic modules can be stored in two ways, which are through a thermophysical reaction, such as a temperature change or phase change material, and through chemical reactions [7]. Effective thermal management of PV systems is the best way to maintain the efficiency of these systems at their highest values [8].

In phase change materials (PCMs), thermal energy is stored during the phase change in a material from solid to liquid or from liquid to gas, and these materials can store energy at a high density in the form of latent heat [9]. The change in the substance phase causes heat to be absorbed or emitted. Latent heat is better than sensible heat because it stores from 5 to 14 times the value of heat per unit volume [10]. During the phase change in PCMs, a nearly constant temperature is maintained during this process. Therefore, adding PCMs to devices that generate large amounts of heat, such as electronic devices, is one of the most important methods adopted in the effective thermal management of these devices for their safe and reliable operation [11].

Phase change materials (PCMs) are divided into the following three sections: inorganic (hydrate, molten salt, metal), eutectic (organic–organic, organic–inorganic, and inorganic–inorganic compounds) and organic (paraffin, stearic acid) [12]. It is reported that the organic PCM has a latent heat from 10 kJ/kg to 300 kJ/kg. There have been recent advances in the storage of latent heat in solar and industrial applications. These PCMs have become a promising way to store energy and reduce carbon dioxide emissions. The advantage of using latent thermal storage is their high storage density, constant temperature, and that the technology can be used more than once. Therefore, it is recommended to use it. However, phase change materials should be packaged so that the liquid phase does not leak out, and to avoid contamination, especially when the PCM contains water to prevent water evaporation or absorption [12].

Large volumes of PCMs can be packed into a large container by microencapsulation technology but, due to the low thermal conductivity of PCMs, the process could fail. Heat transfer is greatly disrupted when the PCM is in the tank with both phases (liquid and solid) in one container. Recently, specifically after 2005, the focus was on integrating latent heat storage into solar energy, as a system was designed with a heat exchanger linked to photovoltaic cells [13].

Paraffin wax is a pure mixture of saturated carbon and hydrogen atoms that forms chemical alkanes of high molecular weight expressed by the chemical formula (CnH2n + 2). It is prepared from petroleum and shale oil; paraffin was known in the 1850s when scientists were able to figure out how to separate waxy substances from petroleum. Paraffin wax can be classified according to its hardness [14,15]. The characteristics of paraffin wax are distinguished by its color, ranging from pure white to dark black, according to the concentration of carbon atoms in its chemical composition. In addition, the melting point and density of paraffin wax are between 16–300 °C and 900 kg/m$^3$, respectively. Furthermore, paraffin wax is soluble in esters, ether, and benzene while it is insoluble in the water. Paraffin has a low burning rate and remains thermoplastic in its solid state at room temperature; meanwhile, it adheres to the surface when exposed to heat [16].

In modern applications, especially in thermoelectric photovoltaic (PV/T) systems, the speed of heat storage and heat disposal has become an important issue [17]. There are many valuable research papers published in the literature about this issue. Different nanomaterial additives of nano-$Al_2O_3$, nano-$ZnO_2$ and nano-SiC, etc., were added to paraffin and the

changes in thermophysical properties of nanomaterials were studied [18]. Therefore, the aim of these tests is to reach the best material suitable for including in PV/T systems in different working conditions.

The results of these studies showed a slight or limited increase in the heat capacity of the nano-PCMs mixtures while the thermal conductivity of the product was increased. Nano-PCM has been used in many solar applications, and studies have shown a significant difference in the rates of improvement in thermal conductivity and product stability. Table 1 shows some of the studies related to adding multiple nanoparticles to paraffin wax to achieve the best heat storage, thermal conductivity, and thermal stability.

**Table 1.** Some studies which used nano-PCMs in variable solar energy applications.

| Ref. No. | Country | Year | PCM Used | Nanoparticles Added | Main Findings |
|---|---|---|---|---|---|
| | | | Solar distillers | | |
| [18] | Saudi Arabia | 2016 | Paraffin wax | Nano-Cu | Nano-copper has been added to paraffin to enhance its low thermal conductivity and distiller performance. The addition of nano-Cu to paraffin enhanced the thermal storage process and increased the yield of distilled water by about 125% compared to the usual distiller. This addition caused an increase in the operating time of the distiller after sunset by 6 h. |
| [19] | India | 2017 | Paraffin wax | Nano-CuO | The addition of nano-CuO to paraffin increased the yield of the distillate by about 35%. The daily productivity of the distiller increased when it was made with the composite material by about 34.7% compared to adding pure paraffin to the distiller. |
| [20] | Iraq | 2020 | Polyvinyl pyrrolidone (PVP K-30), polyethylene glycol (PEG 6000) and carboxymethyl cellulose sodium salt (CMC) | - | The daily distilled yield was increased by 30–50% of the distilled water compared to the distiller without PCM. The working time of the distiller with PCM increased and the distillation continued even after sunset. The quantity of distilled water increased by 120%, while the efficiency of the solar still with PCM increased by about 40%. |
| [21] | India | 2021 | Paraffin wax | Nano-CuO | More distillation was achieved using nano-PCM. The highest distillation yield was when 0.3% by weight was added to paraffin, the yield of the distillate improved by 62.74% compared to the conventional solar still. |
| [22] | India | 2021 | Paraffin wax | Nano-Si | Experiments were carried out in hot and humid atmospheres in Coimbatore, India, during the month of April 2020. The addition of paraffin and nano-Si composite material improved the yield of the distiller by 67.07%. While when using pure paraffin, the productivity improvement was only 51.22%. |
| [23] | Egypt | 2022 | Paraffin wax | Nano-CuO | The use of PCM and CuO nanofluid improved the performance of the modified hemispherical distiller by 80.20%. It also reduced the cost of distilled water by up to 90%. |
| [24] | Saudi Arabia | 2022 | Paraffin wax | Nano-$Al_2O_3$ | The productivity of the composite material (Nano-$Al_2O_3$-paraffin) distiller improved by about 95% compared to the usual distiller. The thermal efficiency of the solar still for this case also increased to 62.4% compared to 30% for the conventional still. |
| | | | Solar air heaters | | |
| [25] | Malaysia | 2016 | Paraffin wax | Nano-Cu | When nano-Cu was added by weight (0.5%, 1.0%, 1.5%, and 2.0%), the thermal conductivity of the composite materials increased by 14.0%, 23.9%, 42.5%, and 46.3%, respectively. The efficiency of the distiller was improved by adding the composite material by 1.7%. |
| [26] | India | 2019 | Paraffin wax | Nano-$SiO_2$ | The energy efficiency of the solar still containing paraffin and nano-$SiO_2$ improved by 74.79%, while, when adding pure paraffin, it was 69.62% compared to the case of conventional distiller with an efficiency of 58.74%. The thermal conductivity of paraffin was increased by 22.78% by mixing it with nano-$SiO_2$. |

**Table 1.** *Cont.*

| Ref. No. | Country | Year | PCM Used | Nanoparticles Added | Main Findings |
|---|---|---|---|---|---|
| [27] | Pakistan | 2019 | RT44HC and RT18HC | - | The study was conducted using high melting point RT44HC paraffin and RT18HC low melting point paraffin filled with circular and semicircular finned tubes. The work of the air heater for the previous two cases lasted about two and three hours, respectively, compared to the non-paraffin heater after sunset. These systems achieved total thermal efficiencies of 68.4%, 71.9%, and 53.2% respectively. |
| [28] | India | 2020 | PCM | MWCNTGNT | The authors designed and manufactured an efficient chemically and thermally stable polyethylene glycol 6000 (FS-PCM) heat sink with an addition of 1 wt.% (MWCNTs/graphene nano-sheets) to it. The results showed a significant improvement in thermal conductivity of 61.73% and 84.48%, respectively. The energy storage properties were improved by using the FS-PCM participation, and the heat sink temperature was reduced by 9.77%. |
| [29] | Iraq | 2021 | Paraffin wax | - | The thermal efficiency of the solar heater containing paraffin increased to 16.3% while, in the case without paraffin, it was 12.4%. The time during which the heater can heat the air increased after sunset. |
| [30] | Egypt | 2021 | Paraffin wax | Nano-CuO | It was found from practical experiments that the best concentration of Nano-CuO added to paraffin is 1%, in order to achieve the best improvement in the phase change (melting) period of about 22.22% compared to paraffin. This addition also reduced the solidification time of the composite material to a high level of 66.66% compared to the hardening time of pure paraffin. |
| [31] | China | 2021 | Paraffin wax | Nano-$Al_2O_3$ | The addition of nanoparticles to paraffin caused an improvement in the thermal conductivity, which escalates by increasing the added mass fraction, reaching an improvement of up to 15% when adding nano-$Al_2O_3$ with a mass fraction of 5%. The dynamic viscosity of paraffin increased by increasing the mass fraction added to it. The experiments proved that the best mass fraction added that gives acceptable results is 1%. |
| [32] | Ethiopia | 2022 | Paraffin wax | Nano-$Al_2O_3$ | The melting rate of paraffin was improved by adding nano-$Al_2O_3$ by 10%, bringing this improvement to 225%. As for the solidification rate, it was improved by 180%. Paraffin's ability to store latent and sensible heat has also been relatively reduced. |
| Domestic solar water heaters | | | | | |
| [33] | Spain | 2009 | 80% Paraffin + 20% stearic acid | Graphene | The use of PCM with graphene caused the water temperature to increase by 3–4 °C within 10–15 min. It also increased the thermal storage efficiency of the heater by 74%. |
| [34] | Jordan | 2010 | Paraffin wax | - | The storage of paraffin in aluminum tubes inside the water heater caused an increase in water temperature by 13–14 °C. The water heater maintained a temperature higher than the surroundings, about 30 °C, as a result of thermal storage. |
| [35] | Tunisia | 2014 | Paraffin wax | - | Two cavities were added to the design of the domestic water heater to store paraffin, which was used to store energy as latent heat. The geyser water stayed hot for 5 h after sunset. Energy efficiency increased up to 35%. |
| [36] | Malaysia | 2014 | Paraffin wax | Nano-Cu | The addition of nano-composite (paraffin and nano-Cu) increased the water retention temperature to 10.8 °C. The solar water heater efficiency has also been increased up to 52.0%. |
| [37] | Malaysia | 2017 | Paraffin wax | - | The different weather conditions of the study area and the rates of flow and withdrawal of hot water showed that the thermal efficiency of the system is about 38–42% on sunny days, while on cloudy days it becomes around 34–36%. |
| [38] | Iran | 2017 | Paraffin wax | Nano-CuO | The addition of nanoparticles to paraffin causes it to reach its melting point faster than in the case of pure paraffin. The temperatures of the upper part of the composite material rise when working with a normal load. |

**Table 1.** *Cont.*

| Ref. No. | Country | Year | PCM Used | Nanoparticles Added | Main Findings |
|---|---|---|---|---|---|
| [39] | India | 2019 | Paraffin wax | Nano-CuO | The addition of nanoparticles to paraffin caused an increase in its thermal conductivity. The high thermal conductivity of nano-CuO caused most of the heat to be transferred to the water with low absorption from paraffin. The maximum temperature obtained is 57.7 °C. The use of the composite material increased the operating time of the heater during the night. |
| [40] | Indonesia | 2019 | Paraffin wax | - | The water was heated to a temperature of 56 °C when the intensity of the solar radiation was 1289 W/m$^2$. The water temperature is affected by the position of the paraffin-filled copper tubes. |
| [41] | India | 2020 | Paraffin wax | Nano-CuO | The addition of Nano-CuO–paraffin to the water heater caused the water temperature to rise to 61.8 °C when the heater was working in a continuous flow, while when the water was stored for 30 min, the water temperature increased to 80.6 °C. The highest heat transfer rate was 5.32 kW in the case of adding a composite material. |
| [42] | India | 2022 | Paraffin wax | Nano-Si | The addition of nano-Si to paraffin hindered its dissolution at high temperatures. The volume fraction of nanoparticles added up to 1% affects the thermal properties as well as the thermal stability of paraffin. |
| PVT systems | | | | | |
| [43] | India | 2019 | Paraffin wax | Nano-graphene | When nano-graphene was added to paraffin with a mass fraction of 3%, the thermal conductivity of the product increased by 146% compared to that of pure paraffin. |
| [44] | India | 2019 | Paraffin wax | Nano-SiO$_2$ | When 2.0% of nano-SiO$_2$ was added to paraffin, its thermal conductivity increased by 33.34% compared to pure paraffin. The latent heat of paraffin decreased with the increasing mass fraction of nano-SiO$_2$. The improvement of the properties of the thermoplastic composite material is evident, especially when nanoparticles are added with a low mass fraction. |
| [45] | Saudi Arabia | 2020 | Paraffin wax RT35 Paraffin wax RT25Eutectic of capric-palmitic acidCalcium chloride hexahydrate CaCl$_2$·6H$_2$O | Nano-CuMWCNTGNTNano-Diamond (Di) | The use of a PCM layer between the collector and the PV plate causes the temperature of the PV panel to increase by 7–27%, while if this layer is not added, the panel's temperature increased by 41%. Calcium chloride hexahydrate CaCl2.6H2O and paraffin RT35 proved to have the best efficiency when nanoparticles were added to them. Additionally, MWCNT was the most effective nanoparticle tested by increasing the cooling of the PV module. |
| [46] | China | 2021 | Paraffin wax | Nano-Al$_2$O$_3$ | The addition of nanoparticles to PCM caused an increase in electrical energy and thermal energy to 136.93 and 377.87 W/m$^2$, respectively, while the energy level decreased to 2.91 W/m$^2$. It was possible to reduce the generated entropy by changing the phase of the PCM at a thickness of 2.75 cm, and by adding nano-Al$_2$O$_3$ with a mass fraction of 8.05% to the paraffin and 8% to the nanofluid. |
| [47] | Egypt | 2022 | paraffin wax RT35 | Nano-Al$_2$O$_3$ | The studied cooling system works by adding paraffin to nano-Al$_2$O$_3$ liquid water with a mass fraction of 0.4%. Circulating this nanofluid and PCM at a rate of 1.6 L/min caused a decrease in the temperature of the PV module by about 12.11 °C compared to a conventional PV. This fluid also increased the electricity generation of the PV panel by a range of 25.33% to 37.81% for a full day of operation compared to the standalone PV panel. The efficiency of the cooled PVT system with this nanofluid and PCM was 82.7%. |
| [48] | Iran | 2022 | Paraffin | Nano-graphene | The rate of heat transfer increased with an increase in the water flow rate through a PVT system that is cooled using a nano-graphite/paraffin composite material. By adding nano-graphite to paraffin with a mass fraction of 1%, the temperature of the PV module in the system was reduced from 63 °C to 37 °C. The electrical efficiency of the upgraded PVT system increased to 21.2%. |

From a review of the above table, the growing interest in using nano-PCMs in energy storage processes, especially solar energy, is obvious. Indeed, nano-PCM technology has demonstrated a great potential in this field and has increased the efficiency of all the applications that it was tested in. Several PCM systems have successfully passed their tests, and many of these materials have been examined with great interest, although paraffin is perhaps the most interesting. Many nanomaterials of different sizes and shapes were used and mixed with PCM to increase its thermal conductivity. Of these materials, the oxides of copper, aluminum, and titanium were the most tested because of their high thermal conductivity, moderate prices, and good mixing with these materials. It is noticeable that nano-$Fe_2O_3$ was used in a limited way in nanofluids. According to the above review, it was found that there was no interest in the use of nano-$Fe_2O_3$ mixed with paraffin for any solar applications. The reason for this is not understood, although these particles have been tested in nanofluids and their merit has been established [49]. Nano-$Fe_2O_3$ has high thermal conductivity, it is available in Iraq's local markets, and its cost is reasonable. Therefore, the objective of this study is to evaluate the rate of enhancements of a nano-$Fe_2O_3$–paraffin compound's thermophysical properties and whether it can be used efficiently in photovoltaic thermal (PV/T) systems under harsh weather conditions. Until today, the scientific society has not agreed on one optimal nano-PCM for PV/T systems operations. Hence, there are many trials to achieve this nano-PCM and the current study aims for the same purpose. The findings from this work will come up with important data that can promote further understanding and assist the designers in choosing the optimal mixture for photovoltaic thermal applications that use the nano-phase change materials.

## 2. Materials and Methods

### 2.1. Study Area

The current study carried out at the University of Technology in the city of Baghdad, the capital of Iraq, which is located at 44 longitude and 33 latitude. In the middle of the city is the Tigris River, which divides it into two parts, namely Karkh and Rusafa. Baghdad is the capital of the Republic of Iraq. The city's population is currently about 9 million, while the number increases in the morning during official working hours to much higher than this due to visitors and workers from other states and cities. Baghdad is considered one of the most densely populated cities in the world, ranking 40th globally, second in the Arab world, after Cairo. When comparing the above numbers with the population census of Iraq in 1977, the republic's population has increased by nearly 13 million. The high expansion of urbanization that came at the expense of agricultural land can be observed. Desertification has also increased in all parts of Iraq, and the rate of rainfall has decreased to the point of interruption for more than two decades, which has caused clear climate change [50]. The climate of the Baghdad city is considered continental and is very hot in summer. The temperatures in the months of July and August in summer in the shade reach 49 °C during the day and increase on a number of days to more than 50 °C, while at night the temperature is not less than 37 °C. The relative humidity rises in July and at the beginning of August to higher than 50% [50]. The solar radiation in these days rises to about 1100 W/m$^2$ at noon [51]. These conditions are considered very harsh for the operation of PV systems, as they cause a serious deterioration in their electrical efficiency. According to all the above-mentioned conditions, the proposed systems were tested during this period. The Iraqi weather is usually dusty most of the time, which lasts for several days, and dust storms must also be considered, which reduce the solar radiation intensity and cause an accumulation of dust and pollutants on the surface of the PV modules [48]. Therefore, for the purpose of neutralizing this factor and not allowing it to interfere with the effect of temperature, clear days were chosen for the measurements, in addition to cleaning the photovoltaic modules before sunrise.

## 2.2. Materials

Figure 1A shows the paraffin that was used as PCM; it was produced by the Al-Doura refinery in Baghdad. The main specifications of paraffin are listed in Table 2, which are supplied by the refinery. The Nano-$Fe_2O_3$ was manufactured by Sky Spring Nanomaterial and purchased from local markets before being used as an additive to paraffin in four mass fractions (0.5%, 1%, 2%, and 3%). These mass fractions were selected because the larger the number of nanoparticles added, the smaller the distances between the particles dispersed in the paraffin. The halving of these distances allows for a greater collision between the molecules during the liquid state of paraffin and then their bonding together, which increases its weight and causes its deposition at the bottom of the container. This condition is expressed by a decrease in the stability of nano-paraffin, which usually happens when adding large mass fractions. Here, nano-$Fe_2O_3$ were selected because it has acceptable price in markets (about 1.5 US$/gram) and mainly because it currently sees very limited use in PV/T systems that contain paraffins. Figure 1B shows the used nanoparticle in its plastic bag. Table 3 lists the nano-$Fe_2O_3$ specifications supplied by the manufacturer. From revising the specifications of both studied materials, it is noted that the thermal conductivity of the selected nanoparticles is about 21 times greater than that of paraffin. In this case, any added nanoparticles mean that, no matter how limited the concentration, if they are spread properly, it will improve the thermal conductivity of the mixture. Additionally, it can be noticed that the density of nano-$Fe_2O_3$ is 5.47 times higher than that of paraffin. For this reason, the amount of added particles should not be excessive, so as not to clearly increase the density of the product. In this case, the experiments were determined with an added mass fraction of not more than 3%.

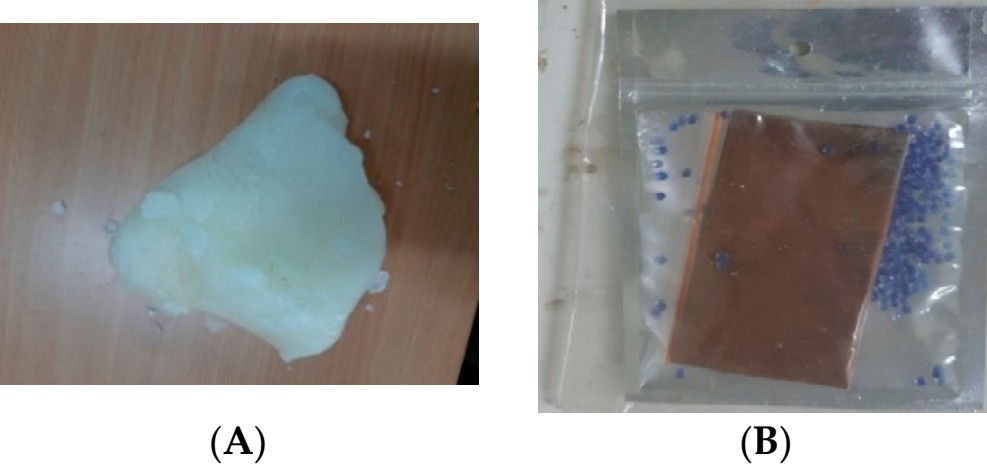

**(A)** **(B)**

**Figure 1.** Photos for the used paraffin and nano-$Fe_2O_3$. (**A**) White paraffin wax. (**B**) Nano-$Fe_2O_3$.

**Table 2.** The studied paraffin's thermophysical properties.

| PCM Type | Paraffin |
|---|---|
| Appearance | White |
| Melting point | 315 K |
| Solidification point | 314 K |
| Density of liquid state | 815 kg/m$^3$ |
| Density of solid state | 910 kg/m$^3$ |
| Latent heat of fusion | 189 kJ/kg |
| Thermal conductivity | 0.21 W/m K |
| Liquid state specific heat | 2.13 kJ/kg |
| Solid state specific heat | 2.22 kJ/kg |

**Table 3.** Nano-$Fe_2O_3$ specifications.

| Material | Nano-$Fe_2O_3$ |
|---|---|
| Manufacturer | Sky spring nanomaterial, Inc. |
| Material appearance | Red brown |
| Purity | 99.5% |
| pH degree | 6 |
| Particles' size | 30–50 nm |
| Density | 5.242 g/$cm^3$ |
| Humidity rate | $\leq$0.056% |
| Molar mass | 159.69 (g/mole) |
| Melting point | 11,838 K |
| Thermal conductivity | 4.66 (W/m K) |

*2.3. PVT System Descriptions*

A container for the nano-paraffin was designed and prepared in a way that allows it to be attached to the back face of the PV panel. This container was built using galvanized iron sheets. Table 4 shows the used container measurements. Paraffin absorbs the accumulated heat in the PV panel in a uniform manner so as to prevent the formation of various heat loads on some areas of the panel. A direct flow heat exchanger of copper tubes was installed inside the container, and this type of heat exchanger was chosen because it draws heat uniformly from the nano-paraffin without leaving uncooled areas inside the container (Figure 2a). The same collector was attached to the back of another PV panel to cool it by circulating water. The paraffin container was insulated with glass wool (25 mm thickness) that covered all exposed sides of it to prevent heat from leaking from the thermal tank to the surrounding air (Figure 2b). Three PV panels, type STF-120P6, were used; the first is standalone, the second is a water-cooled PV/T system, and the third contains the nano-paraffin container and a water circulation exchanger. The PV modules used in this study had a maximum electrical efficiency of 14%, with maximum power ranges from 117 W to 123 W, an open circuit voltage of 20 V, and a short circuit current of 7.63 A. The PV panels were cast facing south at a tilt angle of 33°, based on the results of prior work by Al-Ghezi et al. [47]. In the experiments, two pumps were used to circulate cooling water for the second and third systems. The data acquisition system and a laptop computer were used to monitor and record this data. The practical experiments were conducted outdoors on the roof of the Energy and Renewable Energies Technology Center of the University of Technology, Baghdad. All measurements and data for the three systems were read and continually recorded at the same time. It is worth noting that the energy consumed by the two pumps was not included in the calculations of the performance of the systems.

**Table 4.** Nano-paraffin container measures.

| The Measure | Symbol | Value |
|---|---|---|
| Length | L | 1.145 m |
| Width | b | 0.505 m |
| Perimeter | P | 3.3 m |
| Area | A | 0.578225 $m^2$ |
| Container thickness | x | 0.0275 m |
| Glass wool thermal conductivity | $k_{gw}$ | 0.042 W/$m^2$ K |

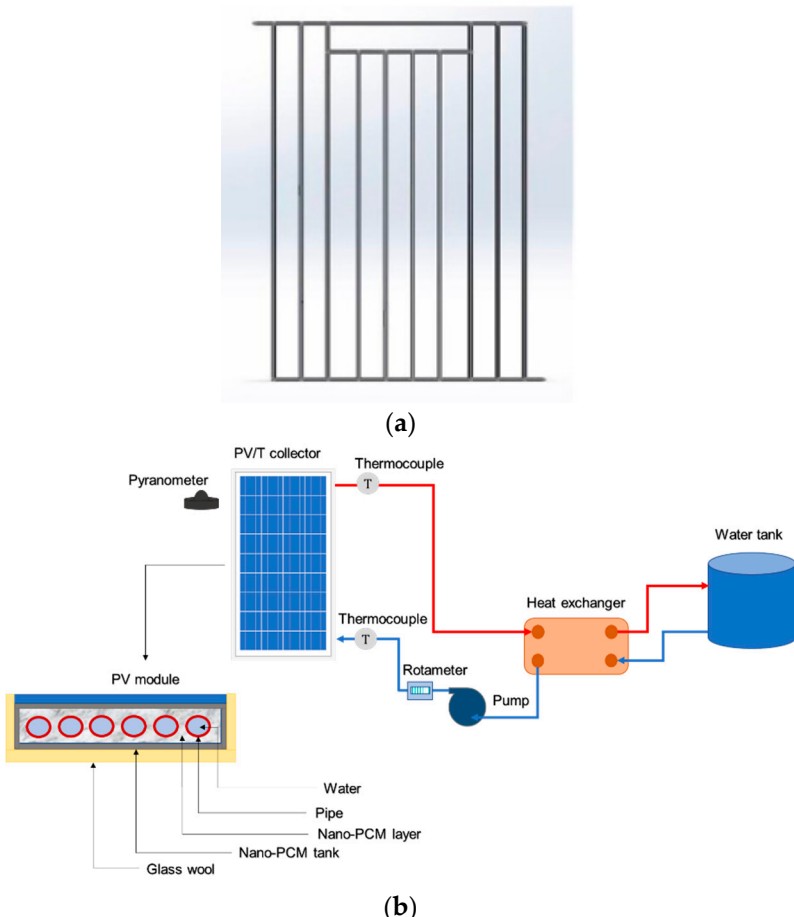

**Figure 2.** The PV/T system with (**a**) direct-flow heat exchanger and (**b**) schematic drawing of PV/T system.

### 2.4. Instrumentations

Several tools were used to accomplish practical experiments for this study, such as a scanning electron microscope (SEM), specifically the VEGA SEM, made in the Czech Republic. It was used to study the outer surface of the nano-paraffin samples.

A vacuum drying oven (made in Hanau, Germany) was used to dry the nanoparticle samples. The oven temperature range is between 70 °C and 200 °C. An ultrasonic bath, type KQ3200E (made in Shanghai, China) was used for mixing samples. This device contains a 12-L bath and an 800 watt heater to keep the samples hot. It also has a frequency of 80 kHz. A Brookfield DV-II+ Pro Viscometer (made in the Middleboro, MA, USA) was used to measure the viscosity of compound materials. This instrument is characterized by ease of preparation and installation, as well as the ease of reading the values through a digital screen. It can measure viscosity even with a small amount of the sample (15 mL). A thermal conductivity meter of the HOT DESK TPS 500 type (made in Göteborg, Sweden) was used to measure thermal conductivity, specific heat, and thermal diffusivity. A sensitive balance, type EJ6I0-E (made in Toruńska, Poland), was used to weigh the materials in four mass fractions. The nano-paraffin density was measured using a density tester (type DII-300 L). The stability of the composite material was measured using a Zetasizer (ZSN) potential analyzer. A thermocouple sensor, type pt100, and a thermometer were used to measure the temperature in variable places, such as the PV panels' surfaces, backs, inner and outer water, and paraffin temperature.

Because the current study is practical, the uncertainty of the different measurements is analyzed to ensure the accuracy of the data through the calibration of the measuring devices. The calibration process is carried out by comparing its measurements with those

of a standard instrument and determining the deviation in the readings [49]. Table 5 lists the measurement equipment and uncertainty for each. The Klein and McClinton equation listed below was used to determine the uncertainty assessment of the measurements [49].

$$W_R = \sqrt{\left(\frac{\partial R}{\partial x_1} w_1\right)^2 + \left(\frac{\partial R}{\partial x_2} w_2\right)^2 + \cdots + \left(\frac{\partial R}{\partial x_n} w_n\right)^2} \qquad (1)$$

**Table 5.** The used measuring instruments and their uncertainties.

| Measured Parameter | Instrument | Uncertainty |
|---|---|---|
| Mass fraction | Sensitive balance (type EJ6I0-E) | ±0.56 |
| Stability | Zetasizer (ZSN) | ±0.62 |
| TC | HOT DESK TPS 500 | ±0.77 |
| Heat Capacity | HOT DESK TPS 500 | ±0.83 |
| Density | Density tester (type DII-300 L) | ±0.49 |
| Viscosity | Brookfield DV-II+ Pro Viscometer | ±1.05 |
| Temperature | Thermocouple sensor (type pt100) | ±1.13 |
| Coolants' Flow rates | HC (US Hunter) | ±0.67 |

The tests uncertainty for the current study is as follows:

$$W_{R_1} = \left[(0.56)^2 + (0.62)^2 + (0.77)^2 + (0.83)^2 + (0.49)^2 + (1.05)^2 + (1.13)^2 + (0.67)^2\right]^{0.5} = 2.24$$

This result shows an uncertainty of less than 5% in the measurements, which means that the accuracy of the conducted measurements is acceptable. The experiments were repeated at least three times to confirm tests repeatability.

### 2.5. Preparation of Nano-Paraffin

The process of preparing nano-paraffin is an important process that has been studied in depth and depends on many variables that must be taken care of in its selection [50,51]. First, it is preferable to heat the nanoparticles in an oven to get rid of moisture and to prevent the particles from clumping during mixing. The nanoparticles' mass fractions were weighed in the sensitive scale and gradually added to the molten paraffin at a temperature at least 10 degrees above the melting point, as indicated in [52]. The sonication process is used to mix and spread nanoparticles in hot paraffin and prevent their agglomeration. The importance of this process is great, and it determines the stability of the paraffin nanoparticles. In this study, the results from the study of Chen et al. [53] were adopted, and three and a half hours was chosen as the sonication time for all mixtures. When the color of the white paraffin is completely changed, the success of the mixing process can be confirmed. In the current experiments, the paraffin color changed to black without impurities (Figure 3). This change means that the nano-$Fe_2O_3$ is uniformly diffused inside the paraffin and bound to its molecules.

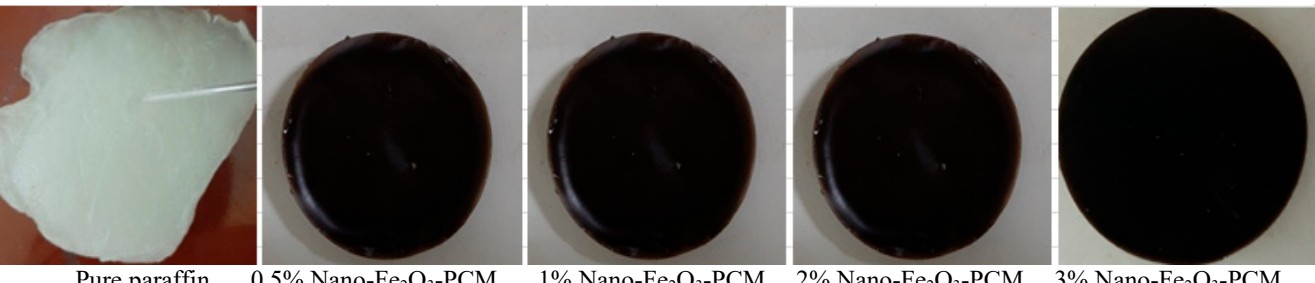

Pure paraffin    0.5% Nano-$Fe_2O_3$-PCM    1% Nano-$Fe_2O_3$-PCM    2% Nano-$Fe_2O_3$-PCM    3% Nano-$Fe_2O_3$-PCM

**Figure 3.** Images of the produced nano-$Fe_2O_3$-paraffins.

## 3. Results and Discussion

### 3.1. Optical Properties

Adding nanoparticles to the paraffin and mixing them well causes a change in the color of the composite material and turns its color to black, as shown in Figure 4. The change in the total color of the nano-paraffin is important evidence that proves the success of the mixing process. However, this color change does not mean that the mixture is stable, so care must be taken on both characteristics, knowing that both depends on the time of sonication and the degree of vibration, as stated by references [1,54].

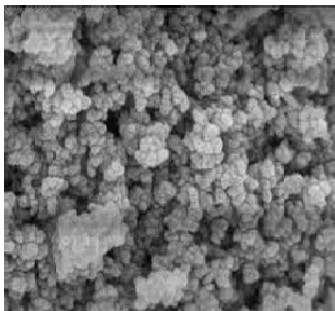

**Figure 4.** SEM test for nano-$Fe_2O_3$.

The scattering and absorption of light for any material expresses its components, and its change means a change in the structural properties of the material. Paraffin consists of chains of hydrogen and carbon with a crystalline structure affected by the interference of nano-$Fe_2O_3$, as the composition of the molecular structures of the mixture changes. The SEM image of nano-$Fe_2O_3$ is shown in Figure 4. This image shows that the nano-$Fe_2O_3$ used is spherical in shape, that it ranges from 30 nm to 50 nm in scale. The shape of the nanoparticle and its measurements greatly affect the thermophysical properties of the nano-paraffin product, as explained by reference [55].

Figure 5 shows the result of the FTIR assay for a nano-paraffin composite material of two nano-$Fe_2O_3$ mass fractions with additions of 1% and 3%. The strong vibration effect appears in the form of sharp peaks as in 3657, 3954, 3158, and 2227, which show that the paraffin structure is vibrating. The absence of a slight peak adjacent to a large peak indicates that the vibration is symmetric. From observing the differences between the two samples, no other peaks appeared, which means that no chemical reaction occurred between nano-$Fe_2O_3$ and paraffin.

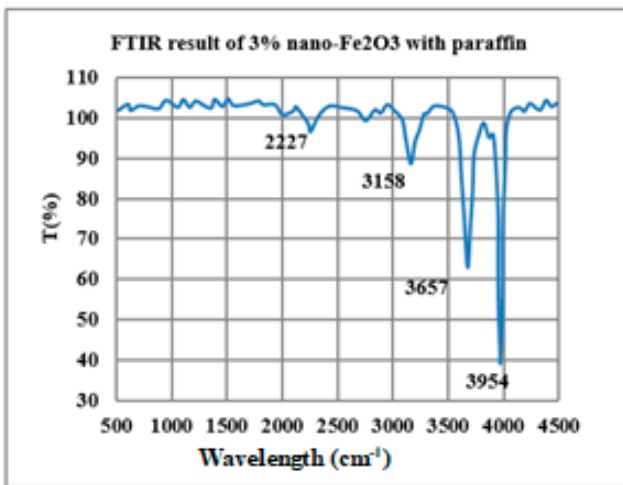 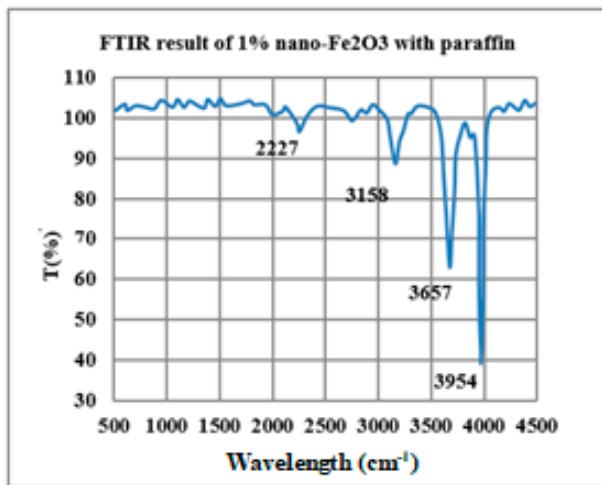

**Figure 5.** FTIR result of nano-$Fe_2O_3$ with paraffin composite material.

### 3.2. Thermophysical Properties

Table 6 lists the measurements of the thermophysical properties of the used paraffin and nano-composites. The main properties measured were density, viscosity, thermal conductivity, and stability.

**Table 6.** Thermophysical measurements for the nano-paraffin composite material for the mass fractions added.

| Temp. (°C) | Paraffin | Nano-PCM (0.5%) | Nano-PCM (1%) | Nano-PCM (2%) | Nano-PCM (3%) |
|---|---|---|---|---|---|
| | | | Density ($kg/m^3$) | | |
| 25 | 910 | 936 | 962 | 1014 | 1067 |
| 35 | 860 | 886 | 912 | 964 | 1016 |
| 45 | 815 | 841 | 867 | 919 | 971 |
| 55 | 780 | 806 | 832 | 884 | 936 |
| 65 | 723 | 749 | 775 | 827 | 879 |
| | | | Viscosity (m Pa·s) | | |
| 25 | 0.093 | 0.095 | 0.096 | 0.097 | 0.098 |
| 35 | 0.062 | 0.065 | 0.066 | 0.068 | 0.069 |
| 45 | 0.053 | 0.055 | 0.056 | 0.057 | 0.058 |
| 55 | 0.49 | 0.053 | 0.055 | 0.058 | 0.059 |
| 65 | 0.042 | 0.044 | 0.046 | 0.047 | 0.049 |
| | | | Thermal conductivity (W/m K) | | |
| 25 | 0.21 | 0.26 | 0.33 | 0.36 | 0.375 |
| 35 | 0.22 | 0.274 | 0.353 | 0.388 | 0.405 |
| 45 | 0.24 | 0.29 | 0.377 | 0.42 | 0.437 |
| 55 | 0.255 | 0.3 | 0.4 | 0.45 | 0.472 |
| 65 | 0.27 | 0.316 | 0.43 | 0.48 | 0.51 |
| | | | Stability | | |
| 25 | - | - | - | - | - |
| 35 | - | - | - | - | - |
| 45 | - | 93 | 87 | 66 | 58 |
| 55 | - | 89 | 84 | 59 | 52 |
| 65 | - | 86 | 79 | 53 | 46 |

### 3.3. Density

Table 5 shows an increase in the density of the composite material compared to paraffin, and this density increases when increasing the addition of the nanomaterial's mass fraction. The density also decreases with increasing the paraffin temperature. Increasing the density of the material is detrimental to applications where there is movement and flow of the fluid. However, in PVT applications, there is no movement of the nano-paraffin in the tank, so increasing its density does not cause harm in this aspect. However, it causes an increase in the weight of the added composite material, which means that it costs the largest amount of work to strengthen the supports carrying the nano-paraffin container. At 25 °C, the increments in nano-paraffins densities were 2.85%, 5.71%, 11.42%, and 17.25% for added mass fractions of 0.5%, 1%, 2%, and 3% compared to pure paraffin, respectively. Additionally, the density increments at 65 °C were 3.6%, 6.8%, 14.38%, and 21.57% for added mass fractions of 0.5%, 1%, 2%, and 3%, respectively, compared to pure paraffin. The increase in the composite material temperature caused a clear decrease in its density, as the density of paraffin decreased by 5.5%, 10.4%, 13.28%, and 20.55% for temperatures of 35 °C, 45 °C, 55 °C, and 65 °C compared to 25 °C density, respectively. When a 3% nano-$Fe_2O_3$ mass fraction was added to paraffin, the decrements in densities were 4.8%, 9.0%, 12.3%, and 17.6% for temperatures 35 °C, 45 °C, 55 °C, and 65 °C, respectively, compared to 25 °C density. The above results explain the reason for the reluctance of most researchers to use nano-$Fe_2O_3$ in PVT systems, as it adds large weights to the system, which increases the costs for the system's supports. However, when using 0.5% or 1% of the mass fraction, the increase in weight will be limited, not exceeding 6%, which means that no more supports

for the system are needed. Certainly, the use of these ratios needs to ensure their thermal conductivity and stability, which we will discuss in the next paragraphs.

### 3.4. Viscosity

Viscosity is the shear forces between fluid layers that resist the movement of the fluid when there is a difference in pressure on both its sides. In PVT systems, paraffin is confined in the container and does not move, so the viscosity effect is limited in this respect. On the other hand, during operation, the phase of paraffin changes from solid to liquid. In the liquid phase, the nanoparticles try to bond due to the Brownian forces, which will cause them to clump and stagnate at the bottom of the container. As a result, the thermal conductivity of the composite material will decrease. However, if the viscosity of the nano-paraffin is high, it will resist the movement of nanoparticles through the fluid layers. The viscosity of the prepared nano-paraffin samples was tested at a temperature range from 25 °C to 65 °C (the rate of temperature change during operation of the PVT system). The viscosity of the samples changes depending on the mass fraction added. At 25 °C, the products' viscosities increased by 2.15%, 3.2%, 4.3%, and 5.37% for added mass fractions of 0.5%, 1%, 2%, and 3%, respectively, compared to paraffin. At 65 °C, the nano-paraffin viscosities were 4.67%, 9.5%, 11.9%, and 16.66% higher than paraffin, for added mass fractions of 0.5%, 1%, 2%, and 3%, respectively. Increasing the nano-paraffin from 25 °C to 65 °C decreased the samples' viscosities by 53.6%, 52%, 51.54%, and 50% while the pure paraffin viscosity reduction rate was 54.8%. The more viscosity deteriorates, the less its resistance to agglomeration of nanoparticles, so the above results indicate a higher stability of composite materials with a small added mass fraction, such as 0.5% and 1%.

### 3.5. Thermal Conductivity

Increasing the conductivity of the composite material is the main factor in adding nanoparticles with high thermal conductivity, such as nano-$Fe_2O_3$. The diffusion of nanoparticles through paraffin improved the product's thermal conductivity, as the more uniform and wider the spread, the better the thermal conductivity. The measurements of this property in Table 6 show an increase in the thermal conductivity of the samples with an increase in the added mass fraction, and the conductivity increases more with an increase in this fraction. At 25 °C, the products' thermal conductivities enhancement rates were 10.04%, 57.14%, 76.19%, and 78.57% for added mass fractions of 0.5%, 1%, 2%, and 3%, respectively. The enhancement rates at 65 °C were 17.03%, 59.26%, 77.78%, and 88.89% for the added mass fractions of 0.5%, 1%, 2% and 3%, respectively. These results showed a linear improvement in the thermal conductivity up to an added mass fraction of 2%, with several decreases in the conductivity improvement. Therefore, choosing a mass fraction equal to or less than 2% is considered wise, as adding larger quantities will not cause a clear higher improvement and will cause an increase in density and higher costs.

The results of the current study were compared with several references in the literature that mixed PCMs with nanoparticles, as shown in Table 7. This comparison may not be fair to any of the references listed, as there is a wide discrepancy between the nanoparticles used, whether in terms of their sizes, shapes, or added amounts. Additionally, the PCMs used varies seriously (even though most researchers used paraffin because of its good storage properties) due to the different densities and melting points. The diversity of nano-PCM preparation methods results in a difference in the resulting thermal conductivity. Overall, such comparisons can give an indication of the accuracy of the work and the acceptability of the results. The results of the table show that adding 1% of nano-$Fe_2O_3$ to paraffin, as prepared in the current study, produced a composite material with high conductivity that can be compared to adding graphene [56] and nano-Cu-$Cu_2O$-CNT [57].

**Table 7.** Comparison between the current study of thermal conductivity enhancement rate and others from the literature.

| Ref. No. | Year | Added PCM | Nanoparticles Added | Added Mass Fraction | Thermal Conductivity Enhancement Rate |
|---|---|---|---|---|---|
| [56] | 2011 | Paraffin | CNF (carbon nano-fiber) | 1% | 21.7 |
| | | Soy wax | CNT | 1% | 5.8 |
| [58] | 2013 | Paraffin | Short-MWCNT | 5% | 29.62 |
| | | | Long-MWCNT | | 14.81 |
| | | | CNFs | | 11.11 |
| | | | CNPs | | 159.25 |
| [57] | 2019 | Paraffin | Cu-Cu$_2$O-CNT | 5% | 58% |
| [59] | 2019 | Paraffin | Polystyrene-Carbon nanotubes (PS-CNT) PolyHIPE foam | - | 62% |
| [60] | 2019 | Erythritol PCM | Graphene | 1% | 53% |
| [61] | 2019 | Pentaerythritol (PE) | Nano-Al$_2$O$_3$ | 0.1% | 20.9% |
| | | | Nano-CuO | | 6.98% |
| | | | Nano-TiO$_2$ | | 14.05% |
| [1] | 2019 | Paraffin | Nano-Al$_2$O$_3$ | 1% | 3.3% |
| | | | Nano-ZnO$_2$ | | 1.8% |
| | | | Nano-SiC | | 4.2% |
| Current study | | Paraffin | Nano-Fe$_2$O$_3$ | 1% | 57.14 |

### 3.6. Products' Stability

The stability of the products means (in the case of the current study) that the thermal conductivity of these composite materials remains high for the longest possible period and that there is no need to empty them from the container and re-mix them. Nanoparticles tend to aggregate and clump and then settle to the bottom of the container, causing a decrease in the conductivity of the composite material and, thus, a decrease in the efficiency of the system as a whole. The melted samples' stabilities were determined using a Zetasizer potential analyzer, as at solid phase there is no motion between particles. The zeta potential was measured at 45 °C (complete melted phase), and showed 93, 87, 66, and 58 for an added mass fraction of 0.5, 1%, 2%, and 3%, respectively. At 65 °C, the measured potentials were 86, 79, 53, and 46 for an added mass fraction of 0.5, 1%, 2%, and 3%, respectively. The zeta potential of the products decreases with an increase in their temperature due to a decrease in their viscosity and, thus, the possibility of agglomeration of nanoparticles is greater. Zeta potential in the range of 50 to 60 indicated good stability, but when the value is higher than 80, the stability is excellent. From the results of Table 6, it is clear that the addition of nano-Fe$_2$O$_3$ with a mass fraction of 0.5% and 1% gives excellent stability to the composite material compared to other mass fractions added.

Table 8 presents some of the stability results of nano-PCM mixtures measured by zeta potential analysis and compared with the current study. As the case mentioned in the previous paragraph, the comparison may not be fair to some studies, as there are many variables that determine the stability of the mixture, such as the type of nano-material added, the size of its particles, the type of surfactant added, its added quantity, the type of studied PCM, etc. The results of the table showed a high stability of the studied nano-Fe$_2$O$_3$–paraffin due to the careful preparation process and the quality of the nanoparticles.

**Table 8.** Comparison between the current study stabilities measurements and others from the literature.

| Ref. No. | Year | Added PCM | Nanoparticles Added | Added Mass Fraction | Zeta Potential (Stability) |
|---|---|---|---|---|---|
| [60] | 2017 | Candeuba wax | Xanthan gum | 0.2% | 22–30 |
| [61] | 2018 | Paraffin | Nano-CuO | 1.5% | 40 |
| [62] | 2018 | Paraffin | Nano-CuO | 2.5% | 40 |
| [63] | 2018 | Paraffin | Silica submicronic | 1% | 60 |
| [64] | 2019 | Paraffin | Nano-$Fe_3O_4$ | 3% | 50 |
| [65] | 2019 | Paraffin | Nano-$TiO_2$ <br> Nano-CuO | 0.5% | 44.5 <br> 45.5 |
| [66] | 2019 | Paraffin | Nano-$Al_2O_3$ | 5% | 60 |
| [67] | 2022 | Paraffin | Nano-TiO <br> Nano-MgO | 1% | 90 <br> 65 |
| [68] | 2022 | Paraffin | CNT <br> CNT/MgO | 3% | 86 <br> 11.1 |
| Current study | | Paraffin | Nano-$Fe_2O_3$ | 1% | 87 |

*3.7. Charging Period*

In thermal storage applications, the charging and discharging processes are very important, as they are the main engine for absorbing heat energy on the one hand and transferring it to the other side. These two process determine the storage time for the potential energy. The faster these two processes work, the better the performance of the application that contains them. Figure 6 shows the effect of nano-additives on the charging time of paraffin when heated. Charging and discharging experiments were carried out in the same nano-paraffin tank. The tank was left in the charged state without water circulating through the paraffin (it was left to heat for 65 min). In the case of discharging, cold water at 25 °C was circulated for 65 min. Figure 6 shows that the paraffin melting process (time during which all paraffin turns molten) is about 25 min, and its melting point is at 43 °C. When adding nanoparticles to paraffin, the melting point decreased partially with an increasing mass fraction of nano-$Fe_2O_3$. It became 43 °C, 42.2 °C, 40.7 °C, and 40.2 °C for added mass fractions of 0.5%, 1%, 2%, and 3%, respectively. Additionally, the charging time was reduced from 25 min for the case of paraffin to become 20, 20, 15, and 15 min for the case of adding a mass fraction of 0.5%, 1%, 2% and 3%, respectively. The charging time of nano-paraffin is affected by the thermal conductivity of the samples and the higher conductivity, the shorter charging period, and the lower the melting point. This result was confirmed by the references [1,69,70]. The phase change period for paraffin takes longer than in the case of nano-paraffin, because the solid layer that faces the heat flow acts as an insulator that prevents heat transfer to the layers that follow when it is heated. In the case of high conductivity by adding nano-$Fe_2O_3$, the charging process accelerates the transfer of heat through the paraffin layers directly and without obstacles. After completing the paraffin and nano-paraffin phase change to liquid, their temperatures increased, which in turn increased the thermal energy stored in the material as sensible heat. Through the phase change operation, the stored energy is latent heat. During the phase change period, the temperature of paraffin and nano-paraffin remains at the melting point until all of the paraffin is completely melted. The phase shift (from solid to liquid) allows nanoparticles to move through the fluid, and this causes its convergence, coalescence, and then precipitation. Therefore, if the distribution of these particles is irregular and the distances between them are close, their accumulation will facilitate a rapid decrease in the stability of nano-paraffin. Is there a guarantee to prevent such a situation? The only guarantee is to take care during the mixing process of paraffin and nanoparticles, and by choosing the type of nanoparticles added and the mass fraction added. The smaller the particles, the better the diffusion process, and the smaller the added quantity, the greater the spacing between the particles, both of which ensure that the stability of the mixture is not lost during the charging and discharging processes. Additionally, during the system operation and the repetition of

charging and discharging operations, if the time period for these two processes is longer than its condition in the first days of operation, this means that the composite material has begun to lose its stability and must be unloaded and re-mixed. During the practical tests, charging and discharging operations occurred more than 70 times, and no clear change was observed in these processes' timings.

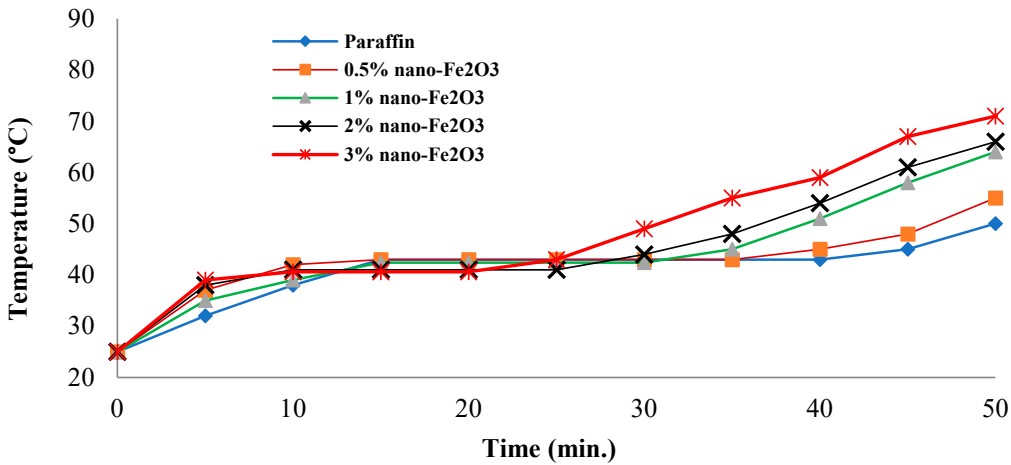

**Figure 6.** Charging period development through heating period.

### 3.8. Discharging Period

During the discharging process, the thermal energy stored in the paraffin is removed, during which the solidification point can be reached, and the phase is changed from liquid to solid. This process is no less important than the charging process. Because of the low thermal conductivity of paraffin, the process takes a long time (25 min), as the temperature of the layer adjacent to the cold surface decreases and acts as an insulator that prevents the flow of heat from the rest of the layers. However, in the presence of nanoparticles, heat transfer continues, so the discharging period is shortened. Figure 7 shows that the addition of nanoparticles significantly reduces the discharging period to 20, 20, 15, and 15 min for added mass fractions of 0.5%, 1%, 2%, and 3%, respectively. Additionally, the solidification point reduced also from 43 °C for paraffin to 43 °C, 42.2 °C, 40.7 °C, and 40.2 °C for added mass fractions of 0.5%, 1%, 2%, and 3%, respectively. The current results are compatible with results from similar studies in the literatures, such as [70].

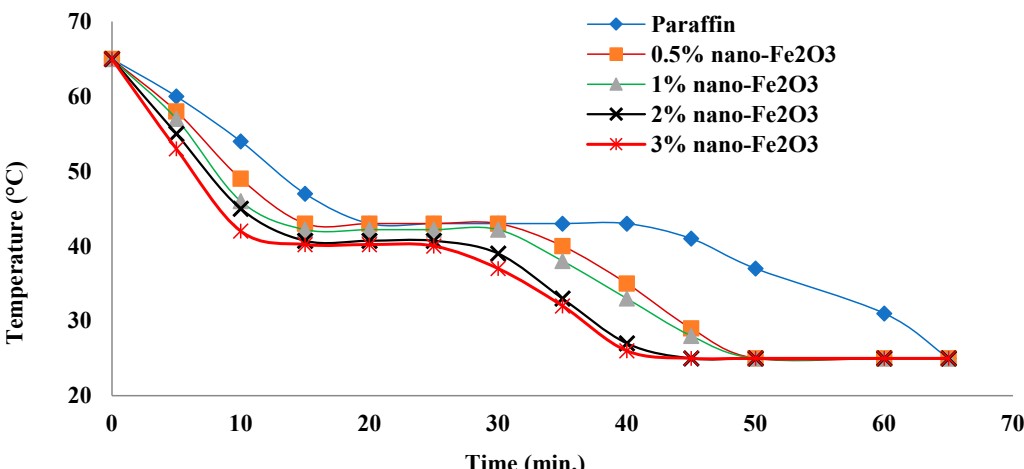

**Figure 7.** Discharging period development during the cooling period.

The results of thermophysical properties are desirable and acceptable for the use of nano-$Fe_2O_3$ and paraffin in PVT systems. The temperature of the photovoltaic panel in Baghdad reaches more than 70 °C, while the paraffin phase changes at a temperature lower than 43 °C. This big difference in the two measurements results in a large heat transfer from the panel to the PCM tank. It can be observed that a good cooling will be achieved with continuous cooling of the nano-paraffin, which is suitable for PVT systems in hot climates. From the available results so far, it is possible to use a mixture of paraffin and nano-$Fe_2O_3$ with a mass fraction of 1%, which is characterized by not consuming a large amount of nano-$Fe_2O_3$ while also raising the thermal conductivity of the product and its excellent stability.

*3.9. PVT System Performance*

Figure 8 shows the relationship between daily operating hours, solar radiation intensity and PV temperatures for the three tested systems during the study period (July and August 2021). This period of the year sees the most severe weather conditions in all of Iraq, and in Baghdad in particular. In addition, the intensity of solar radiation rises to high levels (above 960 W/m$^2$ in the middle of the day) [48]. This radiation intensity is close to measurements made in the sun belt region (Oman [71], UAE [72], and Saudi Arabia [73]). It can be seen that the high radiation intensity leads to an increase in the temperature of the solar panels to high levels. Since the ambient temperatures in the sun are also high, cooling these panels naturally is out of the question, and PVT systems should be used to achieve good cooling. The measurements showed that the average temperature of the standing PV system rises to high levels, reaching more than 80 °C at noon. For a water-cooled PVT system, the temperature of the panel is lower than 50 °C (the maximum PV temperature achieved). For the case of the nano-paraffin PVT system, the temperature of the PV panel was reduced to less than 43 °C (the maximum temperature reached by the PV panel). These measurements confirm the possibility of using PVT systems with nano-paraffin in the cooling of photovoltaic modules. During a whole day operation, the temperature decreased about 20.45% and 34.66% when using water cooling and water-cooled nano-paraffin, respectively. The use of water in cooling the PV modules can be considered efficient (certainly not in the capabilities of the nano-paraffin PVT systems), especially if the costs of the system and the availability of water is included in the calculations. Chaichan et al. [74] proved that in the area of Baghdad, underground water (which is available at levels not exceeding 10 m) can be used to cool PV systems with high efficiency, because the temperature of this water does not exceed 20 °C in summer or winter. For nano-paraffin PVT system, when cooled by water (while the composite material remains at temperatures higher than the solidification temperature) especially during noon times, the cooling efficiency is high. It is noted that the temperature curve for this system is more moderate than the rest of the curves as an indication of high cooling efficiency.

Figure 9 shows the overall competencies of the three studied systems. For a conventional PV system, this system has only electrical efficiency, and this efficiency decreased under the conditions of overheating of PV modules. Therefore, a significant difference is observed between it and the PVT systems. Water is a traditional cooling fluid, so improving the efficiency of the heat exchanger tends to enhance the heat transfer process. In general, the overall efficiency of the nano-paraffin system is very high. A large proportion of heat accumulated in the photovoltaic panel was absorbed by the nano-paraffin cooling system and converted into useful thermal energy. Additionally, this effective cooling of the PV panel led to higher electrical efficiency. The improvement in the total efficiency was about 576.6% and 832. It was 14% for each of the water-based and nano-paraffin PVT systems, respectively, compared to the standalone PV system. The highest total efficiency reached by the nano-paraffin system was 83.43% at two o'clock in the evening, which means that at the most difficult times in terms of weather conditions, the highest solar radiation and ambient temperature increased the efficiency of the system, while in the case of the PV panel, the

efficiency decreased to its lowest levels. According to the results, it can be said that the use of nano-$Fe_2O_3$ added to paraffin is an appropriate and acceptable option.

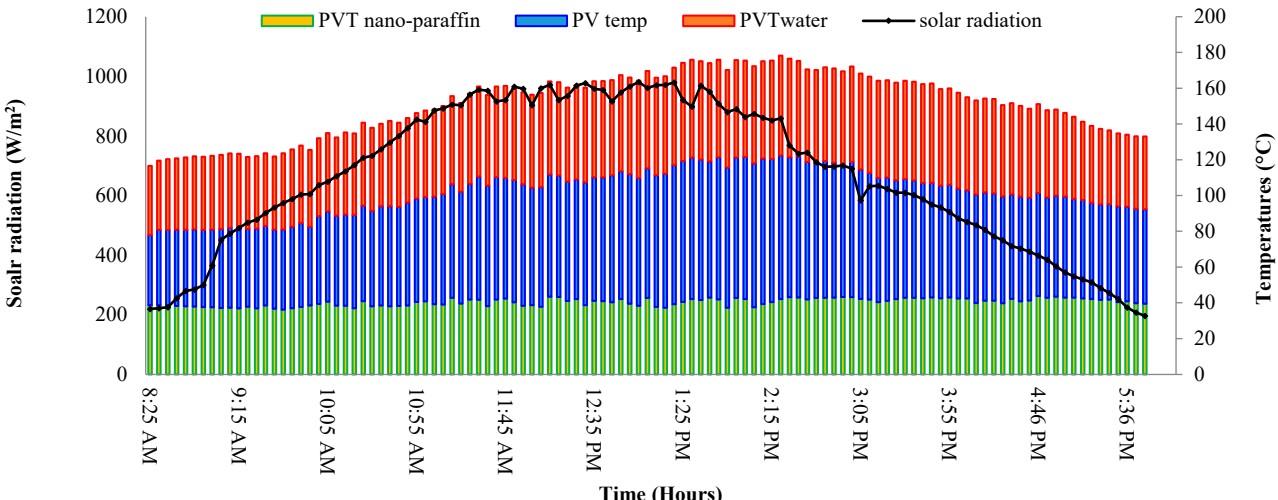

**Figure 8.** Average solar irradiance and systems temperature for the studied period.

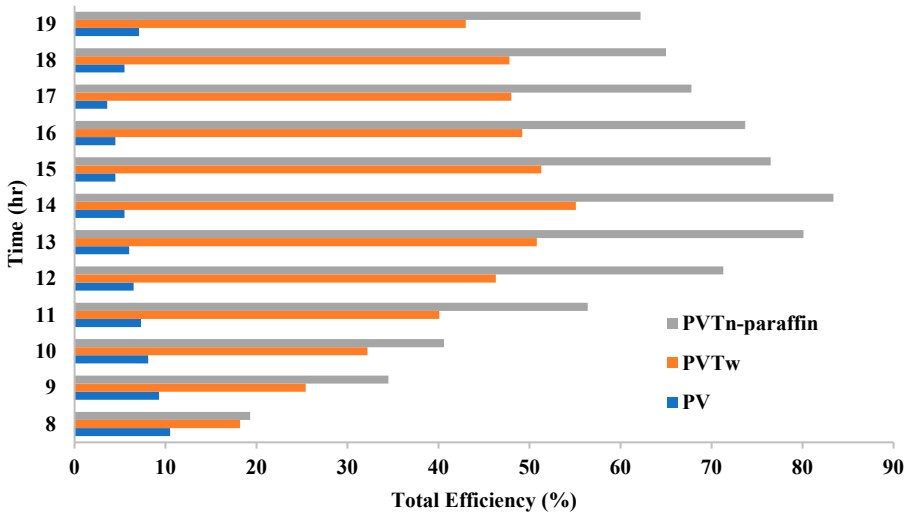

**Figure 9.** The studied systems' total efficiency via day operation time.

In Table 9, the results from the literature showing the differences in the overall efficiency of PVT systems are included. It should be emphasized here that this comparison may be unfair to some references, as it was made for multiple and different PVT systems. In addition to the various types of paraffin used, operating conditions, and the type of nano-material added differed between studies. However, the results from this table can be adopted as an indication of the accuracy and credibility of the results of the current study. It is noted from the results that the nano-$Fe_2O_3$-paraffin system has given an appropriate overall efficiency and is in line with the results obtained by many researchers. This efficiency was achieved despite the system's operation in extremely harsh weather conditions (above 1000 W/m$^2$, 70–80 °C in the sun, and 49–51 °C in the shade). The results of the study showed the possibility of adding nano-$Fe_2O_3$ to paraffin and using them to cool PVT systems successfully in the harsh atmosphere of the city of Baghdad, Iraq.

**Table 9.** Total efficiencies for the tested systems.

| Ref. No. | Year | Added PCM | Nanoparticles Added | Added Mass Fraction | Total Efficiency (%) |
|---|---|---|---|---|---|
| [75] | 2016 | Paraffin | Nano-Ag<br>Nano-Al$_2$O$_3$ | 0.09 | 92<br>86.5 |
| [76] | 2017 | Paraffin | Nano-ZnO | 0.2 | 46 |
| [77] | 2018 | Paraffin | Nano-SiC | 0.5 | 84 |
| [78] | 2019 | Paraffin | Nano-Al$_2$O$_3$ | 1 | 68 |
| [79] | 2020 | Paraffin | MWCNT<br>Nano-MgO | 6 | 61<br>60.6 |
| [80] | 2022 | Paraffin | Nano-Al$_2$O$_3$<br>Nano-Cu<br>Nano-Ag | 4% | 88.64<br>88.71<br>86.63 |
| Current study | | Paraffin | Nano-Fe$_2$O$_3$ | 1% | 83.4 |

## 4. Conclusions

Many researchers have tended to use phase change materials to cool PV modules in PVT systems. Recent research indicated that the addition of nanoparticles with high thermal conductivity could improve their thermophysical properties. In this work, nano-Fe$_2$O$_3$ was added to paraffin with mass fractions of 0%, 0.5%, 1%, 2%, and 3%. It was found that the paraffin viscosity and density increased when Fe$_2$O$_3$ nanoparticles were added. The addition of nano-Fe$_2$O$_3$ improved the thermal conductivity of the product by 30%, 60.5%, 77%, and 86.3% for 0.5%, 1%, 2%, and 3%, respectively. The melting point decreased relatively when increasing the mass fraction of nano-Fe$_2$O$_3$ in the composite material, as well as the phase change period. In the solidification process, the solidification point is reduced by increasing the mass of nano-Fe$_2$O$_3$, as well as the solidification period. The paraffin-nano-Fe$_2$O$_3$ composite material showed high stability when nanoparticles were added with mass fractions of 0.5% and 1%. Practical experiments in the harsh outdoor conditions of Baghdad showed that the nano-paraffin and water-cooled PVT system can reduce the temperature of the PV panel by 34.66%. It also increased the overall efficiency of the system, to reach 83.4%. The overall efficiency of the PVT system in this study was high in comparison with several studies from the literature. The results of the current study confirmed the success of using nano-Fe$_2$O$_3$ added to paraffin in PVT systems operating in very harsh atmospheres.

**Author Contributions:** Conceptualization, M.T.C. and A.A.A.-A.; methodology, M.T.C.; validation, M.T.M., W.N.R.W.I. and M.S.T.; formal analysis, A.A.H.K. and W.N.R.W.I.; investigation, M.S.T.; resources, H.A.K., W.N.R.W.I. and A.H.A.A.-W.; data curation, H.A.K., A.H.A.A.-W. and M.A.F.; writing—original draft preparation, M.T.C. and A.A.A.-A.; writing—review and editing, A.A.H.K., M.A.F. and A.H.A.A.-W.; supervision, M.T.C. and A.A.A.-A.; project administration, H.A.K., A.A.H.K. and A.H.A.A.-W.; funding acquisition, H.A.K., W.N.R.W.I. and M.S.T. All authors have read and agreed to the published version of the manuscript.

**Funding:** This research received no external funding.

**Institutional Review Board Statement:** Not applicable.

**Informed Consent Statement:** Not applicable.

**Data Availability Statement:** Not applicable.

**Acknowledgments:** The authors acknowledge the Universiti Kebangsaan Malaysia (UKM) for their support under research code: GUP-2020-012.

**Conflicts of Interest:** The authors declare no conflict of interest.

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
