# Peer review of "Modified Nano-Fe2O3-Paraffin Wax for Efficient Photovoltaic/Thermal System in Severe Weather Conditions"

_sustainability, doi:10.3390/su141912015_

Round 1
Reviewer 1 Report
The authors of this work have carried out an experimental study on the addition of nano-iron oxide III (Fe2O3) to the paraffin wax, which has been effectively used to improve the thermal and electrical performance of the PVT devices. The results from this study showed that adding Nano Fe2O3 at any mass fraction increases 31 the viscosity and density of the product. Thermal conductivity is improved by adding Nano-Fe2O3 32 to paraffin wax. Stability tests showed that the prepared samples have excellent thermal stability. The nano- Fe2O3 paraffin PV/T system was tested outdoors in the harshest weather conditions of Baghdad city.
In my opinion, the research work is of great interest to the researchers in the field and hence can be published in Sustainability after performing the following minor corrections:
1- What is the main motivation behind the use of Fe2O3 nanoparticles? This should be given in the last section of the introduction.
2- Why the mass fraction of not more than 3% of Fe2O3 was added into the paraffin? How this was calculated?
3- What are the pipe and tank structure materials used in the experimental design?
4- Authors mentioned the uncertainty of 5% for the whole system. Do they mean the fractional% uncertainty or absolute uncertainty?
5- How the authors confirm that the FeO3 particles are not agglomerated during the charging and discharging process?
Author Response
Dear reviewer,
Thank you for useful comments and suggestions
All had been conducted point by point, please see the revised manuscript
Thank you
Best regards

Reviewer 2 Report
This paper experimentally investigates the ferric oxides-based nano-enhanced PCMs for the thermal management of photovoltaics. Although, several studies have been presented and compared in the introduction section, however, the reviewer is not convinced yet. Hence, the novelty of the current work should be further explained and the following points must be addressed before this paper gets accepted for publication in the Sustainability journal.
1. Please review the whole manuscript again to make the language clear to the reader. There are few sentences which show the indistinct meanings
2. There are many similar studies added in the introduction section. So, it is recommended that instead of adding similar studies, there should be various studies based on PCM-based thermal management systems which should cover the various applications for instance electronics thermal management shortly. So, I recommend the following papers to shortly add in the introduction section.
(1) https://doi.org/10.1016/j.ijheatmasstransfer.2022.122591
(2) https://doi.org/10.1016/j.enconman.2020.113466
(3) https://doi.org/10.1016/j.matpr.2021.09.111
(4) https://doi .org/10.1016/j.csite.2019.100543
3. Fig.1 and Fig. 2 should be combined together.
4. Mention the source of the thermophysical properties of the PCMs and the ferric oxide nanoparticles
5. Remove the following sentences from the conclusion section and add them to the introduction section if there is no repetition.
“Solar energy is abundant and free among the renewable energies sources, which is considered one of the most reliable renewable energies. However, there are two obstacles are facing the use of solar energy on the range wide such as energy storage and unavailability of solar energy with absence the sun.”
Author Response

(The authors gave the same response as above.)
